# B Vitamins and One-Carbon Metabolism: Implications in Human Health and Disease

**DOI:** 10.3390/nu12092867

**Published:** 2020-09-19

**Authors:** Peter Lyon, Victoria Strippoli, Byron Fang, Luisa Cimmino

**Affiliations:** 1Department of Biochemistry and Molecular Biology, Miller School of Medicine, University of Miami, Miami, FL 33136, USA; peterlyon@med.miami.edu (P.L.); vfs15@miami.edu (V.S.); b.fang1@umiami.edu (B.F.); 2Sylvester Comprehensive Cancer Center, Miller School of Medicine, University of Miami, Miami, FL 33136, USA

**Keywords:** folate, Vitamin B12, methionine, one-carbon metabolism

## Abstract

Vitamins B9 (folate) and B12 are essential water-soluble vitamins that play a crucial role in the maintenance of one-carbon metabolism: a set of interconnected biochemical pathways driven by folate and methionine to generate methyl groups for use in DNA synthesis, amino acid homeostasis, antioxidant generation, and epigenetic regulation. Dietary deficiencies in B9 and B12, or genetic polymorphisms that influence the activity of enzymes involved in the folate or methionine cycles, are known to cause developmental defects, impair cognitive function, or block normal blood production. Nutritional deficiencies have historically been treated with dietary supplementation or high-dose parenteral administration that can reverse symptoms in the majority of cases. Elevated levels of these vitamins have more recently been shown to correlate with immune dysfunction, cancer, and increased mortality. Therapies that specifically target one-carbon metabolism are therefore currently being explored for the treatment of immune disorders and cancer. In this review, we will highlight recent studies aimed at elucidating the role of folate, B12, and methionine in one-carbon metabolism during normal cellular processes and in the context of disease progression.

## 1. Introduction

The B vitamins comprise a group of eight water-soluble vitamins (B1, B2, B3, B5, B6, B7, B9, and B12) that act as cofactors, precursors, and substrates for numerous biological processes (Table 1). Dietary intake of these B vitamins is essential for the maintenance of human health and deficiencies can have severe health consequences. Almost all of the B vitamins are either directly or tangentially involved in one-carbon metabolism. One-carbon metabolism plays a central role in the generation of methyl donors in the form of S-adenosylmethionine (SAM), the sole methyl donor utilized by DNA, RNA, histone, and protein methyltransferases [1,2,3,4,5]. Methylation is essential for many cellular processes, including protein–protein interactions and epigenetic regulation, which have important roles in embryonic development, cognitive function, and hematopoiesis [6,7]. Furthermore, perturbations in the uptake and homeostasis of B vitamins, which results in deficiency or excess of one-carbon metabolism intermediates, can lead to neurological defects, anemia, aberrant immune responses, and cancer [8,9,10]. Given the central role of vitamin B9 (folate) and vitamin B12 (B12) as direct participants of one-carbon metabolism, this review will focus on their biological roles in human health.

## 2. Key Regulators of One-Carbon Metabolism 

One-carbon metabolism is driven by the folate and methionine cycles that together regulate DNA synthesis and methylation reactions (Figure 1). These two cycles are linked by methionine synthase (MS), the rate-limiting enzyme in one-carbon metabolism that converts homocysteine (Hcy) to methionine (Met) using 5-methyltetrahydrofolate (5-mTHF) as a methyl donor and B12 as an essential cofactor [8,17]. The key regulators of one carbon metabolism are all essential dietary requirements: Methionine, Folate (B9), and Vitamin B12. 

### 2.1. Methionine

Methionine (Met) is one of nine essential amino acids and plays a critical role in multiple biological processes, including the initiation of all eukaryotic peptide synthesis, lipid biosynthesis, and as the primary source of methyl groups utilized by DNA, RNA, histone, and protein methyltransferases [18,19]. Adequate methionine levels are required for normal growth and development and the maintenance of healthy skin, hair, and nails [20]. After protein digestion, methionine is absorbed in the small intestine through sodium-dependent amino acid transporters that are involved in the transport of several amino acids [21]. Methionine is then excreted into the serum through L-type amino acid transporter 4, which also excretes other neutral amino acids [21].

### 2.2. Folate

Folate (Vitamin B9) is an essential micronutrient that is naturally and most commonly found in dark leafy green vegetables. Due to its hydrophilicity, folate and its derivatives depend on highly specific transporters for its absorption in the intestinal tract and other systemic tissues [22]. The reduced folate carrier (RFC) is expressed on most mammalian cells and is the primary mode of transport for reduced folate uptake into systemic tissues at a neutral pH. In addition to RFC, there are folate receptors α and β (FRα/β) that are highly tissue-specific, membrane-bound receptors that bind reduced folates, allowing them to enter cells via endocytosis [23].

### 2.3. Vitamin B12

Vitamin B12 is produced solely by bacteria and archaea through aerobic or anaerobic pathways [24]. Bacteria that synthesize B12 are located mainly in the bodies of higher predatory organisms, thus meat, milk, eggs, and fish are the major sources of dietary B12, with plants and fungi contributing little B12 to the human diet [25]. However, a recent study found that co-fermenting wheat germ with two different bacterial species allowed for B12 fortification, thus creating a mechanism for producing B12 fortified plant-origin products [26]. 

The absorption of B12 from the diet begins with its release from food sources by gastric acid and pepsin in the stomach, followed by its binding to haptocorrin (HC), which is found in saliva and gastric fluid. In the duodenum, B12 is released from HC by pancreatic proteases and is then bound by intrinsic factor (IF). IF is produced by gastric parietal cells in the stomach. This IF-B12 complex binds the cubulin-amnionless receptor in the ileum and is endocytosed [27]. The IF is degraded and B12 is released in the lysosome and enters the cytoplasm. How B12 exits the lysosome is currently unclear, but some groups have hypothesized that LMBD1 and ABCD4 play a role in B12’s export [28,29]. B12 is then released into the blood via the Multidrug resistance protein 1 (MRP1) where it is bound by transcobalamin (TC) or HC. Only holoTC (TC-bound B12) is active and available for uptake by CD320, the B12 receptor expressed on most cells and a member of the LDL receptor family [30].

### 2.4. Co-Dependence of the Folate and Methionine Cycles

The balance of methionine, folate, and B12 from our diet regulates the activity of the folate and methionine cycles, which are mechanistically co-dependent. The folate cycle converts tetrahydrofolate (THF), sourced from dietary folate, into 5,10-methyleneTHF by serine hydroxymethyltransferase (SHMT), an enzyme that requires B6 as a cofactor [31]. Then, 5,10-methyleneTHF acts as a methyl-donor for thymidylate synthase (TS) to synthesize deoxythymidine monophosphate (dTMP) from deoxyuracil monophosphate (dUMP) or for methylenetetrahydrofolate reductase (MTHFR) to generate 5-mTHF [31] (Figure 1).

In the methionine cycle, Met is converted to SAM by methyl-adenosyl transferase 2A (MAT2A) and SAM then acts as a substrate used by a diverse group of methyltransferases. The product of these methylation reactions is S-adenosylhomocysteine (SAH), which can be hydrolysed into homocysteine (Hcy) by S-adenosylhomocysteine hydrolase (AHCY), then resynthesized into Met by MS using 5-mTHF as a methyl donor and B12 as an essential cofactor [8,17,32]. Occasionally, the cobalt group of B12 becomes oxidized to Co(II) during its reaction [33]. Methionine synthase reductase (MTRR) reduces Co(II) back to Co(I), thereby regenerating this essential cofactor for MS activity. Methionine can also be recycled from Hcy using betaine, which is derived from dietary choline, as a methyl donor [7] or directly from SAM through polyamine synthesis and the methionine salvage pathway. Using B6 as a cofactor, Hcy can also be converted to cysteine via transsulfuration for use in glutathione synthesis [34] (Figure 1). 

In addition to the role of B12 as a cofactor in the recycling of Hcy to Met, it also acts as an important cofactor for another metabolic enzyme, methyl-malonyl CoA mutase (MUT). MUT is located in the mitochondria and utilizes B12 in the form of adenosylcobalamin to convert L-methylmalonyl-CoA to succinyl-CoA, which can then enter the tricarboxylic acid (TCA) cycle. Genetic loss of MUT activity can disrupt TCA cycle function [35] and mitochondrial redox function [36,37], suggesting that B12 levels can influence both one-carbon metabolism, energy metabolism, and redox metabolism.

## 3. Methionine, Folate, and B12 Deficiency on One-Carbon Metabolism: Causes and Consequences

### 3.1. Causes of Deficiency

Given the importance of one-carbon metabolism in cell proliferation and survival, it is not surprising that folate and B12 deficiencies can have severe biological consequences. Reduced levels of the micronutrients involved in one-carbon metabolism would logically cause defects in the production of SAM and nucleotide synthesis. Methionine, folate, or B12 deprivation reduces cell proliferation due to blocks in cell cycling [38,39,40,41], and deficiencies in one of these metabolites, either through deprivation or genetic alterations, has been shown to cause global DNA hypomethylation [41,42,43,44]. Methionine deprivation can cause a global decrease in histone methylation [41,45] and models of B12 deficiency have also been shown to decrease the methylation level of proteins [46,47] (Figure 2A).

In normal adults, deficiencies in a sole amino acid like methionine would be rare with adequate nutrition, though a methionine-deficient diet that reduces serum methionine can be achieved in patients [48]. B9 deficiency can occur during pregnancy [49] as pregnant women need to support an immense amount of rapid cell division and tissue growth, and the epigenome also dynamically changes during embryonic development to regulate gene expression at discrete stages of differentiation. Folate deficiency is also associated with malnutrition and lower socioeconomic status.

B12 deficiency presenting with clinical manifestations, hematologic and/or neurologic, is relatively uncommon in the general US population, however, low serum B12 status can be seen in 10.6% of the US population [50]. Global studies of B12 status find much higher rates of deficiency in developing countries [10], possibly due to malnutrition. B12 deficiency is also more prevalent in the elderly, due to a diverse number of factors, such as increased prevalence of pernicious anemia and dietary changes, with rates of deficiency increasing as age advances [10]. B12 deficiency can be the result of a myriad of causes. General malnutrition, chronic alcoholism, and a vegetarian or vegan diet can cause B12 deficiency and these patients are often recommended to take B12 supplements [10]. Maternal deficiency may result in infant deficiency if the baby is breastfeeding [10]. Pernicious anemia, the absence of intrinsic factor (IF), can lead to B12 malabsorption. In fact, B12 was first identified from the liver as a treatment for pernicious anemia [51,52]. The loss of IF can be caused by neutralizing antibodies to IF or autoimmune gastritis, which destroys the parietal cells that produce IF [53]. Gastrointestinal (GI) diseases [10], infection [54,55,56,57], GI surgeries [10], and several medications [58,59,60,61] are all known causes of B12 deficiency. Inhalation of nitrous oxide (N_2_O) as an anesthetic has long been known to cause megaloblastic anemia and bone marrow suppression in patients [62]. Both nitric oxide [63] (NO) and N_2_O [64] inhibits methionine synthase by oxidizing the cobalt group from Co(I) to Co(II). This oxidation of B12 has been shown to specifically inhibit MS and not MUT [65]. 

Polymorphisms in genes related to B12 uptake have also been correlated with decreased serum B12 [66,67,68]. Similarly, polymorphisms in folate uptake genes can cause a folate deficiency [69,70]. Genetic errors in intracellular B12 metabolism have been extensively studied and are excellently reviewed by Froese and Gravel 2010 [71]. These inherited disorders can have varying severity depending on the mutation and affected gene and can be diagnosed shortly after infancy or later in life. Dietary interventions are often recommended in order to improve patient quality of life and early intervention often improves patient outcomes, thus newborn screening for inherited metabolic disorders is common practice in the United States and other countries. While newborn screenings can expose metabolic disorders, infants are not commonly screened for B12 deficiency at birth, though serum B12 screening has gained support in recent years given the severe neurological consequences of infantile B12 deficiency [72,73].

### 3.2. Developmental and Neurological Consequences of B9/B12 Deficiency

Maternal micronutrient deficiencies can have long term repercussions on health of offspring. Both B9 and B12 deficiencies have been linked to adverse pregnancy outcomes, such as low birthweight, preterm delivery, and abruptio placentae [49]. Correlation between low folate or B12 levels and risk of neural tube defects (NTDs) has been well documented [74,75] and periconceptional vitamin supplements containing folic acid (FA) have been shown to significantly reduce NTDs [76,77,78]. FA supplementation reduces NTDs in mothers who had a previously affected pregnancy and in a double-blind prevention trial, oral supplementation with 4 mg of FA per day was shown to reduce recurrent risk of NTDs by 72% [79]. The Centers for Disease Control and Prevention (CDC) in the United States recommends that women who previously experienced NTDs in pregnancy supplement 4 mg of FA per day [80], equivalent to 10 times the recommended daily allowance (400 µg) for women of reproductive years [81]. 

While previous studies have observed correlations between increased serum folate and red blood cell folate levels with a decreased risk of NTDs [78], the mechanism of action of how FA prevents NTDs is not fully known. There have been studies describing several ways in which NTDs are driven by abnormal epigenetic modifications. Proper closure of the neural tube depends on gene silencing induced by DNA methylation and inadequate DNA methyl transferase (DNMT) activity and methylation cycle inhibitors can hinder this process [82]. Since B9 and B12 are linked to both NTDs and one-carbon metabolism, it has been hypothesized that defective one-carbon metabolism is the cause of the NTDs (Figure 2A). Environmental and maternal dietary factors play a crucial role in ensuring normal DNA methylation in the developing embryo, suggesting that folate and B12, key factors in one-carbon metabolism, are instrumental in preventing NTDs through their effects on methylation [82]. Novel discoveries also point to aberrant epigenetic modifications during neural tube development that directly link folate deficiency to NTDs. Inhibiting folate transport and metabolism causes DNA hypomethylation of the *Sox2* locus, leading to neural crest associated defects [83]. In addition, polymorphisms in one-carbon metabolism genes *MTHFR, MTR,* and *MTRR* have been implicated in the risk of developing NTDs [84,85]. However, a recent study found that an alternate mechanism may also be involved in the development of NTDs independently of one-carbon metabolism. Kim et al. found that folate and B12 are competitive antagonists of the aryl hydrocarbon receptor (AhR), a nuclear receptor that binds hydrocarbon compounds, like the poison dioxin, and induces expression not only of detoxifying enzymes, but also of homeostatic and developmental genes [86]. Loss of AhR antagonism by folate or B12 due to deficiency or activation of AhR by known agonists increases AhR activity and mimics symptoms of B9 and B12 deficiency [86]. This research generates a novel causative mechanism for NTDs that could have clinical implications.

B12 deficiency, in addition to NTDs, also affects adult brain function and is associated with other neurological consequences, including myelopathy, neuropathy, and nerve atrophy. These symptoms have been linked to MUT, the other enzyme that relies on B12 as a cofactor. B12 deficiency can lead to defective myelin synthesis due to decreased MUT activity, which causes incorporation of abnormal fatty acids into neuronal lipids [87]. The ensuing demyelination can lead to subacute combined degeneration (SCD) in the spinal cord, resulting in loss of sensation of vibration and proprioception. B12 deficiency has also been reported to cause some neuropsychiatric syndromes, including psychosis, mania, depression, and chronic fatigue syndrome [87]. In healthy elderly individuals and Alzheimer’s patients, low B12 levels have also been correlated with decreased cognitive function [88]. The underlying mechanisms of these neuropsychiatric symptoms associated with B12 deficiency are unclear.

### 3.3. Hematological Consequences of B9/B12 Deficiency

The classical hematological symptom of folate or B12 deficiency is megaloblastic anemia. This anemia is macrocytic, in which the mean corpuscular volume of the erythrocytes is >100 fL and erythrocytes often exhibit anisocytosis. Neutrophils can also have hyper-segmented nuclei and leukopenia and thrombocytopenia may be present. In bone marrow, megaloblastic changes can be observed, with larger erythroblasts with granular nuclei and large granulocyte precursors [89]. This blastic change can even lead to the misdiagnosis of leukemia in B12-deficient patients [90,91] and is most often attributed to a maturation arrest of cells due to impaired DNA synthesis from altered one-carbon metabolism [89]. However, activation of the aryl hydrocarbon receptor, potentially from loss of antagonism due to B12 or folate deficiency, can also cause megaloblastic anemia [86], so this alternate mechanism must not be ruled out and can be explored further in clinical studies.

### 3.4. One-Carbon Metabolism Defiencies and Colorectal Cancer

Disruption of one-carbon metabolism can lead to decreased DNA synthesis, genomic instability, and decreased methyl donor production. Colorectal cancer (CRC) is associated with genomic instability and DNA hypomethylation [92]. Folate and B12 deficiency are also known to increase uracil incorporation into DNA, leading to DNA damage, which can result in the development of CRC [93,94] (Figure 2A). Clinical studies suggest an inverse relationship between folate or methionine intake and CRC risk [93,95,96]. One study found that colonic DNA hypomethylation in patients with CRC correlated with lower folate status, leading to the notion that low folate levels and DNA hypomethylation may be a risk factor in CRC [97]. Another study showed that higher genomic DNA methylation in their leukocytes was associated with a lower risk of colorectal adenomas, which are pre-cancerous lesions, but DNA methylation levels were not associated with folate intake [98]. Both studies provide evidence that hypomethylation could be an important factor in early colorectal carcinogenesis. However, there is not enough conclusive evidence to confirm whether DNA hypomethylation due to folate or B12 deficiency is a direct causal factor in the development of CRC. In addition, it has been shown that 5-azacitidine (5-aza), a DNA hypomethylating agent, has synergistic effects when given with a topoisomerase 1 inhibitor, irinotecan, for the treatment of CRC [99], further complicating the role of DNA methylation in CRC.

### 3.5. Non-Alcoholic Fatty Liver Diseases and One-Carbon Metabolism

The liver plays an important role in the absorption and storage of nutrients and proper liver function is essential for glucose, amino acid, and lipid metabolism. Alterations of dietary nutrients, such as B vitamins, could foreseeably impair proper liver physiology. In fact, altered one-carbon metabolism has been associated with the pathogenesis of nonalcoholic fatty liver disease (NAFLD). NAFLD affects nearly 30% of the population [100] and is characterized by lipid accumulation in the liver, causing increased inflammatory cytokine recruitment, altered resident macrophage function [101], oxidative stress, and even genetic and epigenetic alterations [102,103].

A methionine-choline deficient (MCD) diet has been a well-established model for inducing the full spectrum of NAFLD in rodent studies as methionine, folate, and B12 deficiency all have been linked to NAFLD [104,105] (Figure 2A). MCD mice display altered expression of genes involved in lipid metabolism, as well as elevated SAH and homocysteine levels [104,106] and even impaired mitochondrial function [102]. These studies suggest that disruptions in one-carbon metabolism may trigger the progression of NAFLD to non-alcoholic steatohepatitis (NASH) and eventually liver fibrosis by impairing very low density lipoproteins (VLDL) secretions from the liver [100]. One study in humans found that a choline-deficient diet led to the development of liver dysfunction in as little as 3 weeks [107], while another study showed that choline deficiency was associated with increased fibrosis amongst postmenopausal women [108]. Dietary supplementation of methyl-donors was found to halt progression of NAFLD in mice [109], emphasizing that proper one-carbon metabolism is essential for liver health.

While deficiencies of nutrients essential for one-carbon metabolism may be causing NAFLD due to decreased methyl-donor production or decreased DNA synthesis, other hypotheses are possible. Decreased levels of methyl donors, methionine, and choline could lead to monopolization of B12 by one-carbon metabolism-mediated recycling of methionine to compensate for the lack of these nutrients in the diet. The bias towards methionine recycling would limit the availability of B12 to activate MUT. The MUT enzyme is part of the propionate catabolic pathway, which controls the breakdown of branched-chain amino acids (isoleucine, methionine, threonine, and valine), odd-chain fatty acids, and cholesterol into succinyl-CoA for use by the TCA cycle. Both increased propionyl-CoA, a propionate pathway intermediate, and decreased MUT enzymatic activity have been linked to increased production of odd-chain fatty acid from BCAAs in adipocytes [110]. In fact, one study suggests that B12 deficiency can promote odd-chain fatty acid synthesis through accumulation of methyl-malonic acid, the substrate of MUT [111], and another found that B12 deficiency increases serum cholesterol and triglycerides and increases adiposity in mice [112]. These biological changes may impact the development of NAFLD and underline the importance of both B12 sufficiency and intact one-carbon metabolism for proper liver health.

### 3.6. Fortification and Supplementation for the Treatment of B9/B12 Deficiency

Folate and B12 deficiency have several detrimental effects in children, adults, and developing fetuses. NTDs are often untreatable, so in a preventative effort, the United States Food and Drug Administration (FDA) mandated the fortification of grain and flour products with FA, effective on 1 January 1998 [113]. Over 50 countries now have regulations for mandatory FA fortification, however, not all these programs have been implemented yet [114]. Dietary folates predominantly exist in polyglutamated forms and must undergo hydrolysis to monoglutamated forms for absorption in the jejunum. For this reason, FA is preferred for supplementation, because it does not require this conversion step [115]. After evaluating the impact of fortification on NTDs, specifically on spina bifida and anencephaly, one study found a 19% overall reduction in prevalence [116]. Another study reported a reduction of about 30% in NTDs for Hispanic and non-Hispanic white births [117].

While there is no mandatory fortification of grains and flour with B12, some foods and energy drinks are supplemented with B12. As a treatment for deficiency, B12 supplementation is achieved either via 1 mg oral dosing or 1 mg intramuscular injection. B12 is generally supplemented in the form of cyanocobalamin, but hydroxycobalamin is also used. These forms of B12 are converted intracellularly to the active forms methylcobalamin and adenosylcobalamin for the enzymes MS and MUT, respectively. Intramuscular injection results in much higher B12 serum levels than oral dosing [118], though even in situations of B12 malabsorption, oral supplementation with 1 mg can be sufficient to restore normal B12 levels [89]. 

FA or B12 supplementation can reverse megaloblastic anemia entirely depending on the deficiency, however, for some neurological changes associated with deficiency, the effects can be permanent. Furthermore, for those patients with neurological symptoms associated with deficiency, high doses of FA should be clinically assessed with caution [119] as FA supplementation can mask a B12 deficiency and a risk of long-term neurological damage if lack of folate was not the initial cause [80]. This is potentially due to competition of one-carbon metabolism and MUT for B12: if increased FA causes increased B12 demand for use in one-carbon metabolism, this could disrupt MUT enzymatic function, which is known to cause neurological symptoms. 

## 4. The Role of Methionine, Folate, and B12 Excess in Disease Progression

The role of one-carbon metabolism deficiency has been widely studied in human health, however, the implications of excessive levels of these metabolites is less well known. In the case of folate and B12, there are no reports of direct toxicity of either excess folate or B12 consumption [12], however, there are growing concerns about an upper limit of supplementation. Analyses of FA-fortified foods have reported higher than expected levels of folate compared to the mandated amount by federal regulation and the amount listed on the nutrition labels, some even reaching 300% of reported levels [120]. After the mandatory fortification of foods with folate, a study found that serum levels of folate in the population were double what was predicted, leading to the hypothesis that many people were overconsuming folate-fortified foods [121]. Similarly, the average B12 intake in Americans is almost double the recommended daily intake, though increased intakes are recommended for pregnant women, elderly individuals, and those with potential or known B12 absorption issues [12]. Some multivitamins and energy drinks and supplements contain amounts of B12 greater than 1000 times the daily recommended intake. B12 administration is also available at certain “wellness” clinics around the United States (where intramuscular injections of B12 can cause serum level concentrations that exceed 10X the normal range [118]). A recent study by Arendt and Nexo showed that approximately 15% of hospitalized patients that had B12 measurements taken exhibited high serum levels of B12 [122]. In a separate study of 5771 patients, those with increased serum B12 levels (above 455 pg/mL or 336 pmol/L) had increased all-cause mortality upon follow-up [123]. Whether high B12 serum levels are causative or correlative with morbidity and mortality requires further investigation.

### 4.1. High Serum B9/B12 Levels and Cancer

Altered metabolism is a hallmark of cancer and targeting one-carbon metabolism has recently gained attention for the treatment of various malignancies. Increased folate serum levels may be associated with a higher risk of lung cancer [124], breast cancer [125], and even CRC [126], where low levels are also associated with increased CRC risk. Elevated B12 serum levels have also been associated with an increased risk of cancer within 1 year of follow-up in two large population-based studies [127,128]. Hypercobalaminemia has been observed in both solid and hematological malignancies at diagnosis [129] and is associated with an increased risk of prostate cancer [130] (Figure 2B). 

In the previously mentioned large population studies, elevated serum B12 conveyed the highest risk of hematologic malignancies within 1 year of patient follow-up compared to all other cancers [127,128]. In fact, B12 levels have been known to be increased upon diagnosis in hematological malignancies for over 60 years [131,132,133,134,135]. This may be partially attributed to increased production of HC from cancer cells, leading to increased B12 binding in the serum and decreased cellular uptake [135,136,137]. Increased HC or TC production is also seen in hepatic and pancreatic cancers [138,139,140]. HC expression was recently found to be a negative prognostic biomarker in colon cancer and HC expression decreased after neoadjuvant chemotherapy [141]. Overexpression of HC is also associated with adverse outcomes and poor therapeutic response in rectal cancer [142]. Overall, increased HC levels may be falsely inflating B12 serum levels and could even be depriving tumors and normal tissues of sufficient B12. Future studies assessing B12 levels in cancer patients should include serum holoTC measurements as this is the only protein-bound B12 available for cellular uptake.

Since some cancers are associated with higher levels of serum micronutrients, it is relevant to address the influence of vitamin supplementation on cancer risk. B12 supplement intake is associated with increased lung cancer risk in men (not women), especially in smokers [143,144]. In a randomized trial, FA supplementation was associated with increased prostate cancer risk [145]. Supplementation with B12 and FA for 2–3 years has also been shown to increase the overall risk of cancer and risk of colorectal cancer in a randomized study [146]. This increase in cancer risk may be associated with increased folate-cycle driven metabolism, such as increasing nucleotide synthesis, to the benefit of rapidly dividing cancer cells. One could hypothesize a similar effect for methionine, but a recent meta-analysis comparing high plant protein intake to high animal protein intake found no change in cancer-associated mortality, though there was an increase in all-cause mortality with a high animal protein diet [147].

### 4.2. Excess One-Carbon Metabolites in Immunity and Organ Function

One-carbon metabolism controls DNA synthesis and methyl donor availability, which are essential for the normal function of cells in the immune system. Altered levels of one-carbon micronutrients, B9 and B12, are observed in both normal immune cells and hematological cancers. A study found that un-metabolized FA in the plasma was associated with a decrease in natural killer (NK) cell toxicity, suggesting that excess FA may be associated with immune dysfunction [148] (Figure 2B). Another recent study in mice found that both high and low FA levels impair hematopoiesis, resulting in DNA damage and compromised production of lymphocytes [149]. Rheumatoid arthritis (RA) and other inflammatory diseases have also been associated with hypercobalaminemia, potentially through increased TC in acute inflammation [129]. Conversely, Systemic Lupus Erythmatosus, another inflammatory condition, has recently been linked to lower B12 levels in a meta-analysis [150]. Whether excess of B12 in inflammatory diseases causes immune dysfunction as seen with excess folate requires further investigation.

Preservation of one-carbon cycling is important for solid organ function, as previously noted with the liver and NAFLD. The liver is involved in homeostasis of nutrients like B12, and both the liver and kidney store and excrete B12. Several clinical studies have found increased amounts of serum B12 in patients with liver diseases due to varying mechanisms. Acute hepatitis can lead to the release of stored B12 and decreased TC synthesis [151]. Cirrhosis is thought to cause decreased uptake of HC-bound B12 for excretion in bile and the severity of cirrhosis has been positively correlated to serum B12 levels [151]. Alcoholic liver disease is thought to increase the levels of HC and decrease levels of TC, leading to simultaneously decreased uptake of B12 and increased serum B12 levels [152]. High B12 levels have been associated with reduced kidney function [153], chronic kidney disease [154], and renal failure [155] due to unknown mechanisms. Further studies into the underlying mechanisms and effects of this increased B12 on disease initiation and progression are needed.

## 5. Targeting One-Carbon Metabolism for the Treatment of Disease

While the above research suggests that supplementation with FA and B12 could potentially be harmful, B12 supplementation, even in patients with normal serum B12 levels, has been explored to counteract certain neurological defects, given that B12 deficiency can cause neurological symptoms. B12 treatment does have potential as a therapy for peripheral neuropathic pain [156] and acute herpetic neuralgia [157,158]. This may be due to the restoration of MUT enzymatic activity, which is inversely associated with neuronal damage. However, excessive and prolonged B12 treatment, like in patients with B12 deficiency, can lead to antibodies against TC [159]. These antibodies permit B12 binding to TC, but limit cellular uptake of holoTC [159], ultimately raising serum B12 levels. 

Since adequate B12 is vital for properly-functioning one-carbon metabolism, B12 supplementation has been hypothesized to impact DNA, RNA, and protein methylation. Indeed, administration of B12 has been shown to alter DNA methylation patterns in patients [160]. B12 supplementation has also been shown to promote DNA methylation changes and affect the antidepressant response in a mouse model [161]. Supplementation also restores hippocampal DNA methylation levels in rats with bacterial meningitis [162]. These studies suggest that B12 supplementation could be explored in scenarios where increased methylation potential is required, analogous to the FA supplementation of pregnant women to prevent NTDs.

### 5.1. Folate and Methionine-Cycle Targeted Therapies

One-carbon metabolism has an essential role in nucleic acid synthesis pathways, making it a powerful therapeutic target to slow down cellular proliferation in the treatment of cancer. In fact, some of the first chemotherapies were folate antagonists, known as anti-folates, which act as antiproliferative agents by inhibiting DHFR and disrupting DNA synthesis [163]. The anti-folate aminopterin was the first anti-folate shown to significantly reduce tumor burden in humans [164]. In 1948, the temporary remission of 10 out of 16 children with acute lymphoblastic leukemia (ALL) was observed after treatment with this drug [164]. Since this seminal study, folate antagonists have been used to treat an array of neoplasms, including various types of leukemia, Hodgkin’s disease, lymphosarcoma, breast cancer, and prostate cancer [165]. The most successful anti-folate in chemotherapy to date is methotrexate (MTX), formerly known as a-methopterin. MTX used in high doses remains one of the most effective therapies for treatment of childhood ALL [166]. However, high-dose MTX treatment has been associated with hematological, gastrointestinal, and hepatic toxicity [167,168,169]. Because of these adverse effects, correct dosage and timing is essential in therapy. Further, 5-fluorouracil (5-FU), a uracil analog that inhibits TS, also disrupts the folate cycle and DNA synthesis and is used to treat several types of cancers [170]. Leucovorin, also known as folinic acid, is a derivative of the metabolically active THF that is often administered with MTX as it has been shown to mitigate toxicity and increase efficacy by providing a substrate to restore the folate cycle [169,171]. 

While high-dose MTX is commonly used in cancer treatment, low doses have been shown to be beneficial for the treatment of autoimmune disorders. RA patients treated with low-dose MTX exhibit a significant decrease in the number of tender or painful joints, the duration of morning stiffness, and an overall reduction in disease activity [172]. The mechanism of action of theses anti-inflammatory effects of MTX are not fully understood [173]. Studies have shown that one potential role for MTX in RA treatment could be through its ability to promote demethylation and increased expression of the FOXP3 gene. FOXP3 is an important transcriptional regulator for the differentiation of regulatory T cells, which are known to suppress inflammatory responses in RA patients [174,175]. Being a folate antagonist, MTX acts directly upstream of both nucleic acid synthesis and methionine recycling and therefore has the potential to influence methyltransferase activity and alter the epigenome.

Targeting the methionine cycle within one-carbon metabolism also has demonstrative therapeutic potential. Further, 5-aza and 5-aza-2’deoxycitidine (decitibine) are methylated cytadine analogs that act as DNMT1 inhibitors, which cause DNA hypomethylation and are used in MDS/AML patients. SAM analogs have been developed, which interfere with the enzymatic activity of histone methyltransferases and these drugs may also act as inhibitors of DNMTs and other protein or lipid methyltransferases or affect DNA repair mechanisms [176,177]. Compounds that block the conversion of methionine to SAM are currently being developed in order to disrupt the influence of one-carbon metabolism on the epigenome. However, in light of the myriad of studies describing a link between methionine deficiency and NAFLD, side-effects of interventions targeting the methionine cycle for long-term methionine deprivation should perhaps be approached with caution. In addition, the effects of over-/under-consumption of dietary folate and B12 as it affects one-carbon metabolism targeted therapies, such as 5-azacitidine or 5-FU, should also be explored.

Targeting B12, the cofactor for the rate limiting reaction of the folate- and methionine-cycles, appears an opportune strategy as high serum levels are associated with increased cancer risk. However, it is still not known whether these high serum levels are causative or correlative with disease. Expression levels of MTRR, the gene responsible for reducing B12 to its active form for use by MS, are increased in ovarian carcinoma and targeting MTRR using RNA interference has been shown to inhibit growth and cisplatin resistance in both in vitro and in vivo models, making it a potential target for therapy [178].

### 5.2. Dietary Interventions of One-Carbon Metabolism

Recently, there has been much interest in the influence of diet and lifestyle on human health and disease states. Nutritional deficits of micronutrients can alter cellular metabolism and cause phenotypic changes in humans, thus adjusting the intake of these nutrients could be exploited for therapeutic benefit. Cancer cells are known to have altered metabolism and increasing metabolic stress through nutrient deficiencies could be a suitable addition to current cancer therapies.

Studies in animal models have shown a therapeutic benefit to maintaining a methionine restricted (MR) diet, including an overall extension of lifespan [179,180,181,182]. The earliest reports from the Orentreich group showed that reducing dietary levels of methionine in rats increased longevity by 30–40% [179,181]. Other benefits include improved metabolism [181], decreased adiposity [183,184], and decreased inflammation and oxidative stress [180,185,186,187,188,189,190,191]. 

A reduction in dietary methionine has also been shown to influence the epigenome in mice due to changes in methyltransferase activity [45,192]. Aberrant DNA and histone methylation patterns are a hallmark of cancer [193,194,195] and many genetic and epigenetic functional studies have identified methyltransferases to be frequently mutated in all types of malignancies [196,197,198,199,200,201]. Recent studies highlight a potential therapeutic role for MR in cancer metabolism as many tumors exhibit a higher requirement for exogenous methionine, either due to an increased metabolic need for this amino acid in the maintenance of accelerated tumor growth or due to poor methionine recycling from homocysteine [202,203]. This enhanced metabolic need for methionine in cancer cells can be targeted by restricting methionine availability in the diet or by specifically targeting enzymes and cofactors of the methionine cycle. Sugimura et al. was the first to report anti-tumor effects of an MR diet [204] in 1959 and since then, the field has exploded with different potential interventional avenues. In vitro cell culture models using numerous cancer cell lines or primary patient tumor cells showed that many were intrinsically sensitive to low methionine levels [205,206,207]. Studies using patient-derived xenografts (PDX) models of colorectal cancer [48], prostate cancer [208,209], and pre-malignant breast cancer [210] have all shown that MR also suppressed tumor growth *in vivo.* In a PDX model of colorectal cancer, it was found that the beneficial effect of MR may be through alteration of one-carbon metabolism flux, specifically in regard to redox and nucleotide balance that in combination with antimetabolite or radiation therapy, inhibits tumor growth [48]. MR has also been effective in reducing metastasis in animal models of breast cancer [211] and melanoma [212]. Clinical studies investigating the role of MR in diseases such as cancer are still limited. However, it has been shown that short-term MR in patients with gastric tumors in conjunction with 5-FU has a synergistic effect on reducing TS activity, tumor volume, and weight [213].

While FA fortification has been successful in the prevention of NTDs, studies suggest that both low and high FA may negatively alter immune function. Therefore, consumption of FA should be approached with caution, especially since fortified foods contain more folate than reported levels [120] and people are consuming more than expected [121]. High levels of FA supplementation may be necessary for women of reproductive years, however, monitoring folate levels in blood cell maintenance and malignancy may also be necessary. The influence of folate on the epigenome, specifically DNA methylation in cancer, highlights the need for closely monitoring folate levels in cancer patients. Other than CRC, it is not yet clear whether folate levels can influence cancer initiation or progression in other lineages.

In the case of B12, dietary intervention studies are currently lacking, which warrants investigation. A limited amount of studies have reported no significant reduction in cancer mortality in vegans and vegetarians [214], who likely have decreased B12 serum levels due to lack of meat consumption. However, many vegan and vegetarian individuals take B12 supplements to maintain B12 serum levels. Interestingly, the G allele of a polymorphism in MTRR (A66G) decreases its enzymatic activity and reduces conversion of homocysteine to methionine [215,216], effectively mimicking the effect of reduced B12 availability on one-carbon metabolism. The GG genotype is associated with decreased leukemia risk in Caucasians and children, especially for acute lymphoblastic leukemia [217]. Pooled data from 85 studies show that the homozygosity for the G allele in this MTRR A66G polymorphism is associated with increased overall cancer susceptibility [218]. These genetic studies strengthen the need for more exploration into the effects of B12-deficient diets on cancer outcomes.

## 6. Conclusions and Future Directions

One-carbon metabolism is essential for cellular function and relies on B vitamins to drive and coordinate the generation of methyl groups for a myriad of biological outcomes. Alterations in one-carbon metabolism can be caused by B vitamin deficiencies, leading to developmental, neurological, and hematological consequences. The importance of one-carbon metabolism and the epigenome has been given renewed attention, given that epigenetic dysregulation is a hallmark of many types of cancer, including solid tumors and blood cell malignancies. B vitamins can also influence other diseases that involve aberrant immune responses, as evidenced by the epigenetic block observed in regulatory T cell maturation in patients with arthritis, which can be overcome by inhibiting the folate cycle. The production of methyl donors from the methionine cycle that maintains histone methylation patterns specific to cancer cells also provides the opportunity to target these essential metabolic components specifically for the treatment of cancer [219]. However, the systemic bioavailability of these micronutrients needs to be considered when targeting one-carbon metabolism. For instance, tumor cell-intrinsic effects may be at odds with effects on the tumor microenvironment, as is potentially the case for colon cancer, where DNA hypomethylation is observed in the cancerous colonic epithelia while the surrounding immune cells of these tumors exhibit DNA hypermethylation. In the context of tumor immunology, the levels of these micronutrients could therefore have important implications and complicate treatment strategies. Both dietary and pharmacological interventions should also be explored in future studies to determine the impact of B vitamins and methionine on disease initiation and progression in order to fully understand how one-carbon metabolism plays a role in human health.

## Figures and Tables

**Figure 1 nutrients-12-02867-f001:**
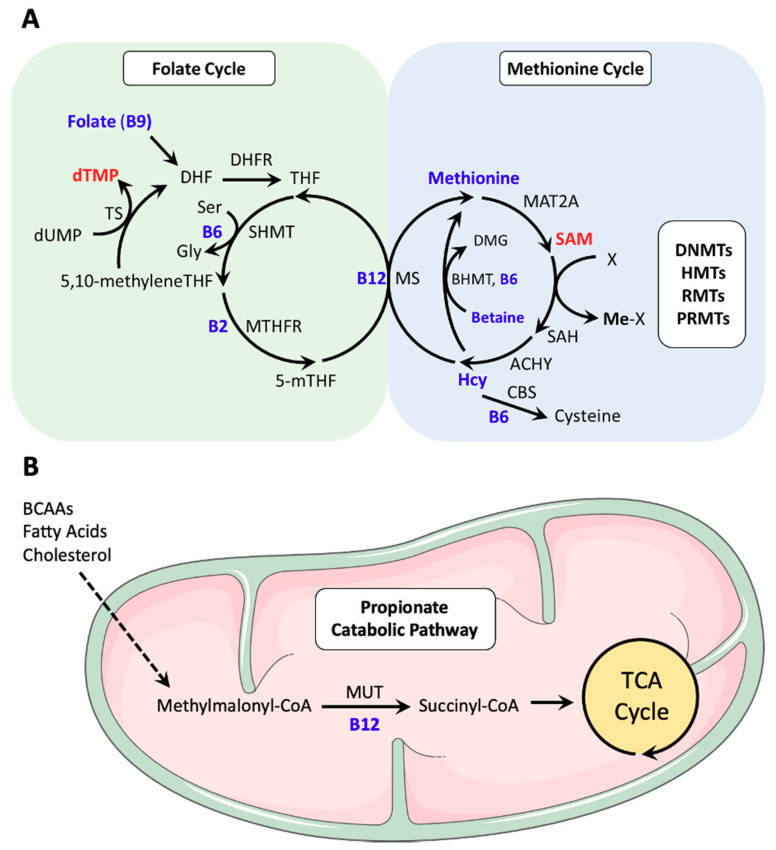
The regulation of metabolism by B vitamins. (**A**) B vitamins in one-carbon metabolism. The folate cycle begins with the conversion of dietary folate (B9) into dihydrofolate (DHF), which is then reduced to tetrahydrofolate (THF) by the enzyme dihydrofolate reductase (DHFR). THF is next converted to 5,10-methyleneTHF by serine hydroxymethyltransferase (SHMT), a reaction that is coupled with the hydroxylation of serine (Ser) to glycine (Gly) and requires B6 as a cofactor. Thymidylate synthase (TS) uses 5,10-methyleneTHF as a methyl donor to methylate deoxyuridine monophosphate (dUMP), creating deoxythymidine monophosphate (dTMP). This step regenerates DHF for continued cycling. Alternatively, 5,10-methyleneTHF can be reduced by methylenetetrahydrofolate reductase (MTHFR) to 5-methytetrahydrofolate (5-mTHF) using B2 as a cofactor. As part of the methionine cycle, 5-mTHF donates a methyl group to regenerate methionine from homocysteine (Hcy), which is catalyzed by methionine synthase (MS) and requires B12, in the form of methylcobalamin, as a cofactor. To generate the methyl donor S-adenosylmethionine (SAM) for use by multiple methyltransferases (MTs) specific for RNA (RMT), DNA (DNMT), histones (HMT), and protein (PRMT) methylation reactions, an adenosine is transferred to methionine by methionine adenosyltransferase 2A. SAM is demethylated during the methyltransferase reactions to form S-adenosylhomocysteine (SAH) that is then hydrolysed by S-adenosylhomocysteine hydrolase (AHCY) to form Hcy. Hcy can also enter the transsulfuration pathway catalyzed by cystathionine beta synthase (CBS) and vitamin B6 to create cysteine. In the liver, betaine from the diet can act as a methyl donor for betaine-homocysteine S-methyltransferase (BHMT), using B6 as a cofactor, to make methionine and dimethylglycine (DMG) as a byproduct. Important dietary micronutrients and metabolite intermediates are highlighted in blue. Items in red are important byproducts of one-carbon metabolism. (**B**) B12 and propionate metabolism. The propionate catabolic pathway breaks down branched-chain amino acids (BCAAs), odd-chain fatty acids, and cholesterol to be used in the tricarboxylic acid (TCA) cycle in the mitochondria. Methylmalonyl-CoA mutase (MUT) converts methylmalonyl-CoA into succinyl-CoA using B12, in the form of adenosylcobalamin, as a cofactor. Succinyl-CoA then enters the TCA cycle.

**Figure 2 nutrients-12-02867-f002:**
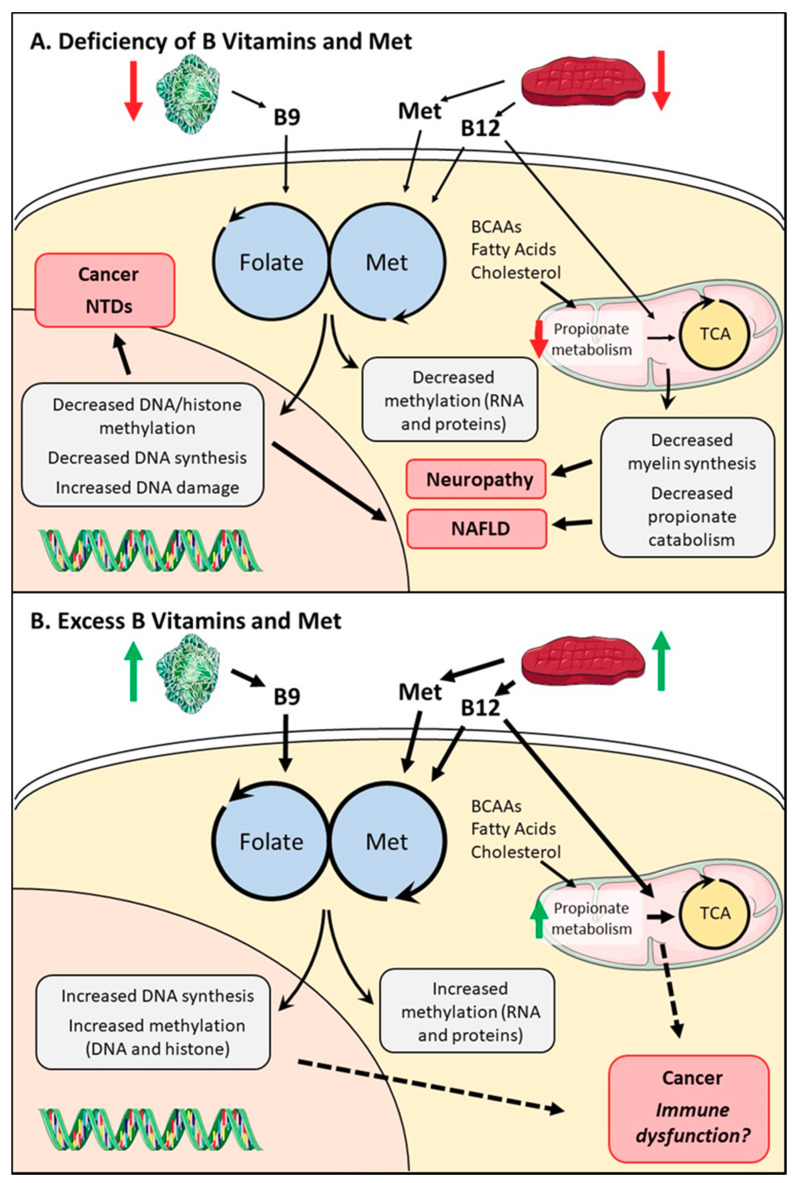
Systemic effects of altered micronutrients in one-carbon metabolism. (**A**) Reduced folate, methionine, or B12 can cause a decrease in one-carbon metabolism output, leading to decreased DNA synthesis, increased genomic instability, and decreased methylation potential. This can promote the development of neural tube defects (NTDs), non-alcoholic fatty liver disease (NAFLD), and cancer (specifically colorectal cancer). Reduced B12 also decreases activity of the propionate catabolic pathway through decreased methylmalonyl-CoA mutase (MUT) enzymatic activity, leading to decreased myelin synthesis, increased cellular stress, and disrupted tricarboxylic acid (TCA) cycling. These factors influence the development of neuropathies and promote NAFLD. (**B**) Effects of excess folate, methionine, and B12 are less understood, but increases can promote cell proliferation and can increase SAM (S-adenosylmethionine) levels, which allow cells to maintain their methylated states. This could lead to the development of cancers as maintenance of methylation is important for some malignancies. Excessive folate also disrupts normal hematopoiesis, possibly through increased one-carbon metabolism.

**Table 1 nutrients-12-02867-t001:** Biological roles of the B vitamins.

Vitamin	Biological Function
B1 (thiamine)	cofactor for enzymes in glucose metabolism, amino acid catabolism, nucleotide synthesis, and fatty acid synthesis [11]
B2 (riboflavin)	precursor for flavin mononucleotide (FMN) and flavin adenine dinucleotide (FAD) for cellular respiration [12]
B3 (nicotinamide)	precursor for nicotinamide adenine dinucleotide (NAD) utilized in biosynthetic pathways, energy metabolism, and protection from reactive oxygen species [13]
B5 (pantothenic acid)	precursor for coenzyme A (coA), an acyl-carrier required for the activity of many enzymes [14]
B6 (pyridoxine)	cofactor for over 150 enzymes involved mainly in amino acid synthesis and degradation [15]
B7 (biotin)	plays an essential role in carboxylation reactions [16] and also has many applications in laboratory research
B9 (folate)	substrate for nucleotide synthesis and methyl-donors in the one-carbon metabolism pathway [12]
B12 (cobalamin)	cofactor for enzymes in one-carbon metabolism and the propionate catabolic pathway [12]

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
