# Peer review of "B Vitamins and One-Carbon Metabolism: Implications in Human Health and Disease"

_nutrients, 2020, doi:10.3390/nu12092867_

Round 1
Reviewer 1 Report
The review manuscript by Lyon et al reviewed the role of micronutrients (folate, B12 and methionine) in one-carbon metabolism in normal and pathological conditions. The authors provide a very good review of different components of one-carbon metabolism, consequences arising from their deficiencies or excess, and targeting one-carbon metabolism for treating different neurological and hematological diseases as well as cancer. The manuscript is very well written with proper illustrations, and therefore, easy to understand. The content of the manuscript is well within the scope of this journal, and it can be very interesting to researchers working in these fields of research.
Reviewer 2 Report
ID: Nutrients-934497
The manuscript is well structured, and concepts are well described and detailed.
Items are properly presented and discussed.
I have only comments on the paragraph 2.4 and 3.1:
Paragraph 2.4
Although the work is mainly focused on one-carbon metabolism by B vitamins, the author should describe the pathway catalised by MUT in Detail, inserting it in Figure 1.
Line 131-132: appropriate literature should be cited for example:
- DOI:3390/ijms21144998
- DOI:3390/ijms19113580
Paragraph 3.1
Line 168-169: Authors should describe the Problems of newborn Screening, when propionic acidemia and methylmalonic acidemia is included, how sensitiv or insensitiv newborn screening for the disorders of cobalamin metabolism is.
Also an appropriate literature should be cited, for example:
- DOI:1016/j.clinbiochem.2014.08.020
Reviewer 3 Report
Interesting review on vitamin B. Perhaps the only issue I think it has not been clearly discussed is the use of demethylating agents in colon cancer and its relation to vitamin B uptake. In general, easy to read and well documented review
